# Making the Cut:
# A Bandit-based Approach to Tiered Interviewing

**Candice Schumann**[⋆]     **Zhi Lang**[⋆]     **Jeffrey S. Foster**[†]     **John P. Dickerson**[⋆]

[⋆]University of Maryland      [†]Tufts University

{schumann,zlang}@cs.umd.edu, jfoster@cs.tufts.edu, john@cs.umd.edu

## Abstract

Given a huge set of applicants, how should a firm allocate sequential resume screenings, phone interviews, and in-person site visits? In a tiered interview process, later stages (e.g., in-person visits) are more informative, but also more expensive than earlier stages (e.g., resume screenings). Using accepted hiring models and the concept of structured interviews, a best practice in human resources, we cast tiered hiring as a combinatorial pure exploration (CPE) problem in the stochastic multi-armed bandit setting. The goal is to select a subset of arms (in our case, applicants) with some combinatorial structure. We present new algorithms in both the probably approximately correct (PAC) and fixed-budget settings that select a near-optimal cohort with provable guarantees. We show via simulations on real data from one of the largest US-based computer science graduate programs that our algorithms make better hiring decisions or use less budget than the status quo.

*'... nothing we do is more important than hiring and developing people. At the end of the day, you bet on people, not on strategies." – Lawrence Bossidy,* The CEO as Coach *(1995)*

## 1   Introduction

Hiring workers is expensive and lengthy. The average cost-per-hire in the United States is \$4,129 [Society for Human Resource Management, 2016], and with over five million hires per month on average, total annual hiring cost in the United States tops hundreds of billions of dollars [United States Bureau of Labor Statistics, 2018]. In the past decade, the average length of the hiring process has doubled to nearly one month [Chamberlain, 2017]. At every stage, firms expend resources to learn more about each applicant's true quality, and choose to either cut that applicant or continue interviewing with the intention of offering employment.

In this paper, we address the problem of a firm hiring a *cohort* of multiple workers, each with unknown true utility, over multiple stages of structured interviews. We operate under the assumption that a firm is willing to spend an increasing amount of resources—e.g., money or time—on applicants as they advance to later stages of interviews. Thus, the firm is motivated to aggressively "pare down" the applicant pool at every stage, culling low-quality workers so that resources are better spent in more costly later stages. This concept of tiered hiring can be extended to crowdsourcing or finding a cohort of trusted workers. At each successive stage, crowdsourced workers are given harder tasks.

Using techniques from the multi-armed bandit (MAB) and submodular optimization literature, we present two new algorithms—in the probably approximately correct (PAC) (§3) and fixed-budget settings (§4)—and prove upper bounds that select a near-optimal cohort in this restricted setting. We explore those bounds in simulation and show that the restricted setting is not necessary in practice (§5). Then, using real data from admissions to a large US-based computer science Ph.D. program, we show that our algorithms yield *better* hiring decisions at equivalent cost to the status quo—or *comparable* hiring decisions at lower cost (§5).

## 2 A Formal Model of Tiered Interviewing

In this section, we provide a brief overview of related work, give necessary background for our model, and then formally define our general multi-stage combinatorial MAB problem. Each of our $n$ applicants is an arm $a$ in the full set of arms $A$. Our goal is to select $K < n$ arms that maximize some objective $w$ using a maximization oracle. We split up the review/interview process into $m$ stages, such that each stage $i \in [m]$ has per-interview information gain $s_i$, cost $j_i$, and number of required arms $K_i$ (representing the size of the "short list" of applicants who proceed to the next round). We want to solve this problem using either a confidence constraint $(\delta, \epsilon)$, or a budget constraint over each stage $(T_i)$. We rigorously define each of these inputs below.

**Multi-armed bandits.** The multi-armed bandit problem allows for modeling resource allocation during sequential decision making. Bubeck *et al.* [2012] provide a general overview of historic research in this field. In a MAB setting there is a set of $n$ arms $A$. Each arm $a \in A$ has a true utility $u(a) \in [0, 1]$, which is unknown. When an arm $a \in A$ is pulled, a reward is pulled from a distribution with mean $u(a)$ and a $\sigma$-sub-Gaussian tail. These pulls give an empirical estimate $\hat{u}(a)$ of the underlying utility, and an uncertainty bound $rad(a)$ around the empirical estimate, i.e., $\hat{u}(a) - rad(a) < u(a) < \hat{u}(a) + rad(a)$ with some probability $\delta$. Once arm $a$ is pulled (e.g, an application is reviewed or an interview is performed), $\hat{u}(a)$ and $rad(a)$ are updated.

**Top-K and subsets.** Traditionally, MAB problems focus on selecting a single best (i.e., highest utility) arm. Recently, MAB formulations have been proposed that select an optimal subset of $K$ arms. Bubeck *et al.* [2013] propose a budgeted algorithm (SAR) that successively accepts and rejects arms. We build on work by Chen *et al.* [2014], which generalizes SAR to a setting with a combinatorial objective. They also outline a fixed-confidence version of the combinatorial MAB problem. In the Chen *et al.* [2014] formulation, the overall goal is to choose an optimal cohort $M^*$ from a decision class $\mathcal{M}$. In this work, we use decision class $\mathcal{M}_K(A) = \{M \subseteq A \mid |M| = K\}$. A cohort is optimal if it maximizes a linear objective function $w : \mathbb{R}^n \times \mathcal{M}_K(A) \to \mathbb{R}$. Chen *et al.* [2014] rely on a maximization oracle, as do we, defined as

$$Oracle_K(\hat{\mathbf{u}}, A) = \underset{M \in \mathcal{M}_K(A)}{\operatorname{argmax}} w(\hat{\mathbf{u}}, M). \tag{1}$$

Chen *et al.* [2014] define a *gap score* for each arm $a$ in the optimal cohort $M^*$, which is the difference in combinatorial utility between $M^*$ and the best cohort *without* arm $a$. For each arm $a$ not in the optimal set $M^*$, the gap score is the difference in combinatorial utility between $M^*$ and the best set *with* arm $a$. Formally, for any arm $a \in A$, the gap score $\Delta_a$ is defined as

$$\Delta_a = \begin{cases} w(M^*) - \max_{\{M \mid M \in \mathcal{M}_K \wedge a \in M\}} w(M), & \text{if } a \notin M^* \\ w(M^*) - \max_{\{M \mid M \in \mathcal{M}_K \wedge a \notin M\}} w(M), & \text{if } a \in M^*. \end{cases} \tag{2}$$

Using this gap score we estimate the hardness of a problem as the sum of inverse squared gaps:

$$\mathbf{H} = \sum_{a \in A} \Delta_a^{-2} \tag{3}$$

This helps determine how easy it is to differentiate between arms at the border of accept/reject.

**Objectives.** Cao *et al.* [2015] tighten the bounds of Chen *et al.* [2014] where the objective function is Top-K, defined as

$$w_{\text{TOP}}(\mathbf{u}, M) = \sum_{a \in M} u(a). \tag{4}$$

In this setting the objective is to pick the $K$ arms with the highest utility. Jun *et al.* [2016] look at the Top-K MAB problem with batch arm pulls, and Singla *et al.* [2015a] look at the Top-K problem from a crowdsourcing point of view.

In this paper, we explore a different type of objective that balances both individual utility and the diversity of the set of arms returned. Research has shown that a more diverse workforce produces better products and increases productivity [Desrochers, 2001; Hunt *et al.*, 2015]. Thus, such an objective is of interest to our application of hiring workers. In the document summarization setting, Lin and Bilmes [2011] introduced a *submodular* diversity function where the arms are partitioned into $q$ disjoint groups $P_1, \cdots, P_q$:

$$w_{\text{DIV}}(\mathbf{u}, M) = \sum_{i=0}^{q} \sqrt{\sum_{a \in P_i \cap M} u(a)}. \tag{5}$$

Nemhauser *et al.* [1978] prove theoretical bounds for the simple greedy algorithm that selects a set that maximizes a submodular, monotone function. Krause and Golovin [2014] overview submodular optimization in general. Singla *et al.* [2015b] propose an algorithm for maximizing an unknown function, and Ashkan *et al.* [2015] introduce a greedy algorithm that optimally solves the problem of diversification if that diversity function is submodular and monotone. Radlinski *et al.* [2008] learn a diverse ranking from behavior patterns of different users by using multiple MAB instances. Yue and Guestrin [2011] introduce the linear submodular bandits problem to select diverse sets of content while optimizing for a class of feature-rich submodular utility models. Each of these papers uses submodularity to promote some notion of diversity. Using this as motivation, we empirically show that we can hire a diverse cohort of workers (Section 5).

**Variable costs.** In many real-world settings, there are different ways to gather information, each of which vary in cost and effectiveness. Ding *et al.* [2013] looked at a regret minimization MAB problem in which, when an arm is pulled, a random reward is received and a random cost is taken from the budget. Xia *et al.* [2016] extend this work to a batch arm pull setting. Jain *et al.* [2014] use MABs with variable rewards and costs to solve a crowdsourcing problem. While we also assume non-unit costs and rewards, our setting is different than each of these, in that we actively choose how much to spend on each arm pull.

Interviews allow firms to compare applicants. *Structured interviews* treat each applicant the same by following the same questions and scoring strategy, allowing for meaningful cross-applicant comparison. A substantial body of research shows that structured interviews serve as better predictors of job success and reduce bias across applicants when compared to traditional methods [Harris, 1989; Posthuma *et al.*, 2002]. As decision-making becomes more data-driven, firms look to demonstrate a link between hiring criteria and applicant success—and increasingly adopt structured interview processes [Kent and McCarthy, 2016; Levashina *et al.*, 2014].

Motivated by the structured interview paradigm, Schumann *et al.* [2019] introduced a concept of "weak" and "strong" pulls in the Strong Weak Arm Pull (SWAP) algorithm. SWAP probabilistically chooses to strongly or weakly pull an arm. Inspired by that work we associate pulls with a cost $j \geq 1$ and information gain $s \geq j$, where a pull receives a reward pulled from a distribution with a $\frac{\sigma}{\sqrt{s}}$-sub-Gaussian tail, but incurs cost $j$. Information gain $s$ relates to the confidence of accept/reject from an interview vs review. As stages get more expensive, the estimates of utility become more precise - the estimate comes with a distribution with a lower variance. In practice, a resume review may make a candidate seem much stronger than they are, or a badly written resume could severely underestimate their abilities. However, in-person interviews give better estimates. A strong arm pull with information gain $s$ is equivalent to $s$ weak pulls because it is equivalent to pulling from a distribution with a $\sigma/\sqrt{s}$ sub-Gaussian tail - it is equivalent to getting a (probably) closer estimate. Schumann *et al.* [2019] only allow for two types of arm pulls and they do not account for the structure of current tiered hiring frameworks; nevertheless, in Section 5, we extend (as best we can) their model to our setting and compare as part of our experimental testbed.

**Generalizing to multiple stages.** This paper, to our knowledge, gives the first computational formalization of tiered structured interviewing. We build on hiring models from the behavioral science literature [Vardarlier *et al.*, 2014; Breaugh and Starke, 2000] in which the hiring process starts at recruitment and follows several stages, concluding with successful hiring. We model these $m$ successive stages as having an increased cost—in-person interviews cost more than phone interviews, which in turn cost more than simple résumé screenings—but return additional information via the score given to an applicant. For each stage $i \in [m]$ the user defines a cost $j_i$ and an information gain $s_i$ for the type of pull (type of interview) being used in that stage. During each stage, $K_i$ arms move on to the next stage (we cut off $K_{i-1} - K_i$ arms), where $n = K_0 > K_1 > \cdots > K_{m-1} > K_m = K$. The user must therefore define $K_i$ for each $i \in [m-1]$. The arms chosen to move on to the next stage are denoted as $A_m \subset A_{m-1} \subset \cdots \subset A_1 \subset A_0 = A$.

**Tiered MAB and interviewing stages.** Our formulation was initially motivated by the graduate admissions system run at our university. Here, at every stage, it is possible for *multiple* independent reviewers to look at an applicant. Indeed, our admissions committee strives to hit at least two written reviews per application package, before potentially considering one or more Skype/Hangouts calls with a potential applicant. (In our data, for instance, some applicants received up to 6 independent reviews per stage.)

While motivated by academic admissions, we believe our model is of broad interest to industry as well. For example, in the tech industry, it is common to allocate more (or fewer) 30-minute one-on-one interviews on a visit day, and/or multiple pre-visit programming screening teleconference calls. Similarly, in management consulting Hunt *et al.* [2015], it is common to repeatedly give independent "case study" interviews to borderline candidates.

## 3 Probably Approximately Correct Hiring

In this section, we present Cutting Arms using a Combinatorial Oracle (CACO), the first of two multi-stage algorithms for selecting a cohort of arms with provable guarantees. CACO is a probably approximately correct (PAC) [Haussler and Warmuth, 1993] algorithm that performs interviews over $m$ stages, for a user-supplied parameter $m$, before returning a final subset of $K$ arms.

Algorithm 1 provides pseudocode for CACO. The algorithm requires several user-supplied parameters in addition to the standard PAC-style confidence parameters ($\delta$ - confidence probability, $\epsilon$ - error), including the total number of stages $m$; pairs $(s_i, j_i)$ for each stage $i \in [m]$ representing the information gain $s_i$ and cost $j_i$ associated with each arm pull; the number $K_i$ of arms to remain at the end of each stage $i \in [m]$; and a maximization oracle. After each stage $i$ is complete, CACO removes all but $K_i$ arms. The algorithm tracks these "active" arms, denoted by $A_{i-1}$ for each stage $i$, the total cost $Cost$ that accumulates over time when pulling arms, and per-arm $a$ information such as empirical utility $\hat{u}(a)$ and total information gain $T(a)$. For example, if arm $a$ has been pulled once in stage 1 and twice in stage 2, then $T(a) = s_1 + 2s_2$.

---

**Algorithm 1** Cutting Arms using a Combinatorial Oracle (CACO)

**Require:** Confidence $\delta \in (0, 1)$; Error $\epsilon \in (0, 1)$; $Oracle$; number of stages $m$; $(s_i, j_i, K_i)$ for each stage $i$

1: $A_0 \leftarrow A$
2: **for** stage $i = 1, \ldots, m$ **do**
3:     Pull each $a \in A_{i-1}$ once using the given $s_i, j_i$ pair
4:     Update empirical means $\hat{\mathbf{u}}$
5:     $Cost \leftarrow Cost + K_{i-1} \cdot j_i$
6:     **for** $t = 1, 2, \ldots$ **do**
7:         $A_i \leftarrow Oracle_{K_i}(\hat{\mathbf{u}})$
8:         **for** $a \in A_{i-1}$ **do**
9:             $rad(a) \leftarrow \sigma\sqrt{\frac{2 \log(4|A| \ Cost^3)/\delta}{T(a)}}$
10:            **if** $a \in A_i$ **then** $\tilde{u}(a) \leftarrow \hat{u}(a) - rad_t(a)$
11:            **else** $\tilde{u}(a) \leftarrow \hat{u}(a) + rad(a)$
12:         $\tilde{A}_i \leftarrow Oracle_{K_i}(\tilde{\mathbf{u}})$
13:         **if** $|w(\tilde{A}_i) - w(A_i)| < \epsilon$ **then break**
14:         $p \leftarrow \arg\max_{a \in (\tilde{A}_i \setminus A_i) \cup (A_i \setminus \tilde{A}_i)} rad(a)$
15:         Pull arm $p$ using the given $s_i, j_i$ pair
16:         Update $\hat{u}(p)$ with the observed reward
17:         $T(p) \leftarrow T(p) + s_i$
18:         $Cost \leftarrow Cost + j_i$
19: $Out \leftarrow A_m$; **return** $Out$

---

CACO begins with all arms active (line 1). Each stage $i$ starts by pulling each active arm once using the given $(s_i, j_i)$ pair to initialize or update empirical utilities (line 3). It then pulls arms until a confidence level is triggered, removes all but $K_i$ arms, and continues to the next stage (line 13).

In a stage $i$, CACO proceeds in rounds indexed by $t$. In each round, the algorithm first finds a set $A_i$ of size $K_i$ using the maximization oracle and the current empirical means $\hat{u}$ (line 7). Then, given a confidence radius (line 9), it computes pessimistic estimates $\tilde{u}(a)$ of the true utilities of each arm $a$ and uses the oracle to find a set of arms $\tilde{A}_i$ under these pessimistic assumptions (lines 10–12). If those two sets are "close enough" ($\epsilon$ away), CACO proceeds to the next stage (line 13). Otherwise, across all arms $a$ in the symmetric difference between $A_i$ and $\tilde{A}_i$, the arm $p$ with the most uncertainty over its true utility—determined via $rad(a)$—

is pulled (line 14). At the end of the last stage $m$, CACO returns a final set of $K$ active arms that approximately maximizes an objective function (line 19).

We prove a bound on CACO in Theorem 1. As a special case of this theorem, when only a single stage of interviewing is desired, and as $\epsilon \to 0$, then Algorithm 1 reduces to Chen *et al.* [2014]'s CLUCB, and our bound then reduces to their upper bound for CLUCB. This bound provides insights into the trade-offs of $Cost$, information gain $s$, problem hardness **H** (Equation 3), and shortlist size $K_i$. Given the $Cost$ and information gain $s$ parameters Theorem 1 provides a tighter bound than those for CLUCB.

**Theorem 1.** *Given any $\delta \in (0,1)$, any $\epsilon \in (0,1)$, any decision classes $\mathcal{M}_i \subseteq 2^{[n]}$ for each stage $i \in [m]$, any linear function $w$, and any expected rewards $u \in \mathbb{R}^n$, assume that the reward distribution $\varphi_a$ for each arm $a \in [n]$ has mean $u(a)$ with a $\sigma$-sub-Gaussian tail. Let $M_i^* = \text{argmax}_{M \in \mathcal{M}_i}$ denote the optimal set in stage $i \in [m]$. Set $rad_t(a) = \sigma\sqrt{2\log(\frac{4K_{i-1} \, Cost_{i,t}^3}{\delta})/T_{i,t}(a)}$ for all $t > 0$ and $a \in [n]$. Then, with probability at least $1 - \delta$, the CACO algorithm (Algorithm 1) returns the set* Out *where $w(\text{Out}) - w(M_m^*) < \epsilon$ and*

$$T \leq O\left(\sigma^2 \sum_{i \in [m]} \left(\frac{j_i}{s_i}\left(\sum_{a \in A_{i-1}} \min\left\{\frac{1}{\Delta_a^2}, \frac{K_i^2}{\epsilon^2}\right\}\right)\log\left(\frac{\sigma^2 j_i^4}{s_i \delta}\sum_{a \in A_{i-1}}\min\left\{\frac{1}{\Delta_a^2}, \frac{K_i^2}{\epsilon^2}\right\}\right)\right)\right).$$

Theorem 1 gives a bound relative to problem-specific parameters such as the gap scores $\Delta_a$ (Equation 2), inter-stage cohort sizes $K_i$, and so on. Figure 1[1] lends intuition as to how CACO changes with respect to these inputs, in terms of problem hardness (defined in Eq. 3). When a problem is easy (gap scores $\Delta_a$ are large and hardness **H** becomes small), the min parts of the bound are dominated by gap scores $\Delta_a$, and there is a smooth increase in total cost. When the problem gets harder (gap scores $\Delta_a$ are small and hardness **H** becomes large), the mins are dominated by $K_i^2/\epsilon^2$ and the cost is noisy but bounded below. When $\epsilon$ or $\delta$ increases, the lower bounds of the noisy section decrease—with the

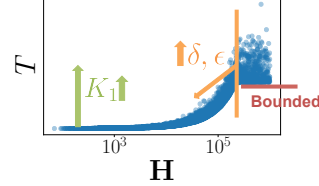

Figure 1: Hardness (**H**) vs theoretical cost ($T$) as user-specified parameters to the CACO algorithm change.

impact of $\epsilon$ dominating that of $\delta$. A policymaker can use these high-level trade-offs to determine hiring mechanism parameters. For example, assume there are two interview stages. As the number $K_1$ of applicants who pass the first interview stage increases, so too does total cost $T$. However, if $K_1$ is too small (here, very close to the final cohort size $K$), then the cost also increases.

## 4 Hiring on a Fixed Budget with BRUTAS

In many hiring situations, a firm or committee has a fixed budget for hiring (number of phone interviews, total dollars to spend on hosting, and so on). With that in mind, in this section, we present Budgeted Rounds Updated Targets Successively (BRUTAS), a tiered-interviewing algorithm in the fixed-budget setting.

Algorithm 2 provides pseudocode for BRUTAS, which takes as input fixed budgets $\bar{T}_i$ for each stage $i \in [m]$, where $\sum_{i \in [m]} \bar{T}_i = \bar{T}$, the total budget. In this version of the tiered-interview problem, we also know how many decisions—whether to accept or reject an arm—we need to make in each stage. This is slightly different than in the CACO setting (§3), where we need to remove all but $K_i$ arms at the conclusion of each stage $i$. We make this change to align with the CSAR setting of Chen *et al.* [2014], which BRUTAS generalizes. In this setting, let $\tilde{K}_i$ represent how many decisions we need to make at stage $i \in [m]$; thus, $\sum_{i \in [m]} \tilde{K}_i = n$. The $\tilde{K}_i$s are independent of $K$, the final number of arms we want to accept, except that the total number of accept decisions across all $\tilde{K}$ must sum to $K$.

The budgeted setting uses a constrained oracle $COracle : \mathbb{R}^n \times 2^{[n]} \times 2^{[n]} \to \mathcal{M} \cup \{\perp\}$ defined as

$$COracle(\hat{\mathbf{u}}, A, B) = \underset{\{M \in \mathcal{M}_\mathcal{K} \mid A \subseteq M \, \wedge \, B \cap M = \emptyset\}}{\text{argmax}} w(\hat{\mathbf{u}}, M),$$

where $A$ is the set of arms that have been accepted and $B$ is the set of arms that have been rejected.

In each stage $i \in [m]$, BRUTAS starts by collecting the accept and reject sets from the previous stage. It then proceeds through $\tilde{K}_i$ rounds, indexed by $t$, and selects a single arm to place in the accept set $A$ or the reject set $B$. In a round $t$, it first pulls each *active* arm—arms not in $A$ or $B$—a total of $\tilde{T}_{i,t} - \tilde{T}_{i,t-1}$ times using the appropriate $s_i$ and $j_i$ values. $\tilde{T}_{i,t}$ is set according to Line 6; note that $\tilde{T}_{i,0} = 0$. Once all the empirical means for each active arm have been updated, the constrained oracle is run to find the empirical best set $M_{i,t}$ (Line 9). For each active arm $a$, a new pessimistic set $\tilde{M}_{i,t,a}$ is found (Lines 11–15). $a$ is placed in the accept set $A$ if $a$ is not in $M_{i,t}$, or in the reject set $B$ if $a$

is in $M_{i,t}$. This is done to calculate the gap that arm $a$ creates (Equation 2). The arm $p_{i,t}$ with the largest gap is selected and placed in the accept set $A$ if $p_{i,t}$ was included in $M_{i,t}$, or placed in the reject set $B$ otherwise (Lines 16-20). Once all rounds are complete, the final accept set $A$ is returned.

Theorem 2, provides an lower bound on the confidence that BRUTAS returns the optimal set. Note that if there is only a single stage, then Algorithm 2 reduces to Chen *et al.* [2014]'s CSAR algorithm, and our Theorem 2 reduces to their upper bound for CSAR. Again Theorem 2 provides tighter bounds than those for CSAR given the parameters for information gain $s_b$ and arm pull cost $j_b$.

**Theorem 2.** *Given any $\bar{T}_i$s such that $\sum_{i \in [m]} \bar{T}_i = \bar{T} > n$, any decision class $\mathcal{M}_K \subseteq 2^{[n]}$, any linear function $w$, and any true expected rewards $\mathbf{u} \in \mathbb{R}^n$, assume that reward distribution $\varphi_a$ for each arm $a \in [n]$ has mean $u(a)$ with a $\sigma$-sub-Gaussian tail. Let $\Delta_{(1)}, \ldots, \Delta_{(n)}$ be a permutation of $\Delta_1, \ldots, \Delta_n$ (defined in Eq. 2) such that $\Delta_{(1)} \leq \ldots \leq \Delta_{(n)}$. Define $\tilde{H} \triangleq \max_{i \in [n]} i\Delta_{(i)}^{-2}$. Then, Algorithm 2 uses at most $\bar{T}_i$ samples per stage $i \in [m]$ and outputs a solution $\mathtt{Out} \in \mathcal{M}_K \cup \{\bot\}$ such that*

$$\Pr[\mathtt{Out} \neq M_*] \leq n^2 \exp\left( -\frac{\sum_{b=1}^{m} s_b(\bar{T}_b - \tilde{K}_b)/(j_b \widetilde{\log}(\tilde{K}_b))}{72\sigma^2 \tilde{H}} \right) \tag{6}$$

*where $\widetilde{\log}(n) \triangleq \sum_{i=1}^{n} i^{-1}$, and $M_* = \operatorname{argmax}_{M \in \mathcal{M}_K} w(M)$.*

---

**Algorithm 2** Budgeted Rounds Updated Targets Successively (BRUTAS)

---

**Require:** Budgets $\bar{T}_i \, \forall i \in [m]$; $(s_i, j_i, \tilde{K}_i)$ for each stage $i$; constrained oracle $COracle$
1: Define $\widetilde{\log}(n) \triangleq \sum_{i=1}^{n} \frac{1}{i}$
2: $A_{0,1} \leftarrow \varnothing$; $B_{0,1} \leftarrow \varnothing$
3: **for** stage $i = 1, \ldots, m$ **do**
4:     $A_{i,1} \leftarrow A_{i-1,\tilde{K}_{i-1}+1}$; $B_{i,1} \leftarrow B_{i-1,\tilde{K}_{i-1}+1}$; $\tilde{T}_{i,0} \leftarrow 0$
5:     **for** $t = 1, \ldots, \tilde{K}_i$ **do**
6:         $\tilde{T}_{i,t} \leftarrow \left\lceil \frac{\bar{T}_i - (n - \sum_{a=0}^{i-1} \tilde{K}_i)}{\widetilde{\log}(n - \sum_{a=0}^{i-1} \tilde{K}_i)j_i(\tilde{K}_i - t + 1)} \right\rceil$
7:         **foreach** $a \in [n] \setminus (A_{i,t} \cup B_{i,t})$ **do**
8:             Pull $a$ $(\tilde{T}_{i,t} - \tilde{T}_{i,t-1})$ times; update $\hat{u}_{i,t}(a)$
9:         $M_{i,t} \leftarrow COracle(\hat{\mathbf{u}}_{i,t}, A_{i,t}, B_{i,t})$
10:         **if** $M_{i,t} = \bot$ **then return** $\bot$
11:         **foreach** $a \in [n] \setminus (A_{i,t} \cup B_{i,t})$ **do**
12:             **if** $a \in M_{i,t}$ **then**
13:                 $\tilde{M}_{i,t,a} \leftarrow COracle(\hat{\mathbf{w}}_{i,t}, A_{i,t}, B_{i,t} \cup \{a\})$
14:             **else**
15:                 $\tilde{M}_{i,t,a} \leftarrow COracle(\hat{\mathbf{w}}_{i,t}, A_{i,t} \cup \{a\}, B_{i,t})$
16:         $p_{i,t} \leftarrow \operatorname{argmax}_{a \in [n] \setminus (A_{i,t} \cup B_{i,t})} w(M_{i,t}) - w(\tilde{M}_{i,t,a})$
17:         **if** $p_{i,t} \in M_t$ **then**
18:             $A_{i,t+1} \leftarrow A_{i,t} \cup \{p_{i,t}\}$; $B_{i,t+1} \leftarrow B_{i,t}$
19:         **else**
20:             $A_{i,t+1} \leftarrow A_{i,t}$; $B_{i,t+1} \leftarrow B_{i,t} \cup \{p_{i,t}\}$
21: $\mathtt{Out} \leftarrow A_{m,\tilde{K}_m+1}$; **return** $\mathtt{Out}$

---

When setting the budget for each stage, a policymaker should ensure there is sufficient budget for the number of arms in each stage $i$, and for the given exogenous cost values $j_i$ associated with interviewing at that stage. There is also a balance between the number of decisions that must be made in a given stage $i$ and the ratio $\frac{s_i}{j_i}$ of interview information gain and cost. Intuitively, giving higher budget to stages with a higher $\frac{s_i}{j_i}$ ratio makes sense—but one also would not want to make *all* accept/reject decisions in those stages, since more decisions corresponds to lower confidence. Generally, arms with high gap scores $\Delta_a$ are accepted/rejected in the earlier stages, while arms with low gap scores $\Delta_a$ are accepted/rejected in the later stages. The policy maker should look at past decisions to estimate gap scores $\Delta_a$ (Equation 2) and hardness $\mathbf{H}$ (Equation 3). There is a clear trade-off between information gain and cost. If the policy maker assumes (based on past data) that the gap scores will be high (it is easy to differentiate between applicants) then the lower stages should have a high $K_i$, and a budget to match the relevant cost $j_i$. If the gap scores are all low (it is hard to differentiate between applicants) then more decisions should be made in the higher, more expensive stages. By looking at the ratio of small gap scores to high gap scores, or by bucketing gap scores, a policy maker will be able to set each $K_i$.

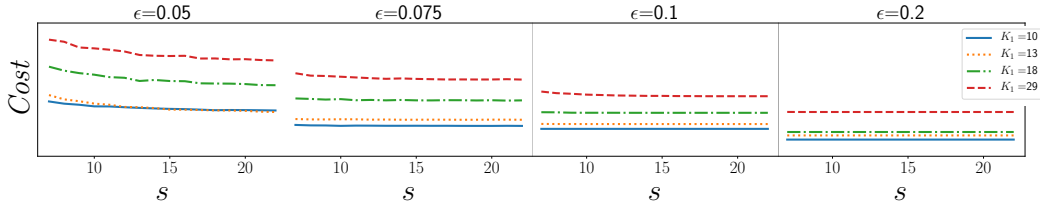

Figure 2: Comparison of $Cost$ vs information gain ($s$) as $\epsilon$ increases for CACO. Here, $\delta = 0.05$ and $\sigma = 0.2$. As $\epsilon$ increases, the cost of the algorithm also decreases. If the overall cost of the algorithm is low, then increasing $s$ (while keeping $j$ constant) provides diminishing returns.

## 5   Experiments

In this section, we experimentally evaluate BRUTAS and CACO in two different settings. The first setting uses data from a toy problem of Gaussian distributed arms. The second setting uses real admissions data from one of the largest US-based graduate computer science programs.

### 5.1   Gaussian Arm Experiments

We begin by using simulated data to test the tightness of our theoretical bounds. To do so, we instantiate a cohort of $n = 50$ arms whose true utilities, $u_a$, are sampled from a normal distribution. We aim to select a final cohort of size $K = 7$. When an arm is pulled during a stage with cost $j$ and information gain $s$, the algorithm is charged a cost of $j$ and a reward is pulled from a distribution with mean $u_a$ and standard deviation of $\sigma/\sqrt{s}$. For simplicity, we present results in the setting of $m = 2$ stages.

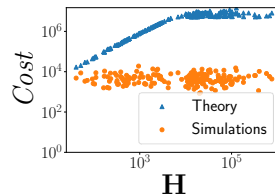

Figure 3: Hardness (**H**) vs Cost, comparing against Theorem 1.

**CACO.**   To evaluate CACO, we vary $\delta$, $\epsilon$, $\sigma$, $K_1$, and $s_2$. We find that as $\delta$ increases, both cost and utility decrease, as expected. Similarly, Figure 2 shows that as $\epsilon$ increases, both cost and utility decrease. Higher values of $\sigma$ increase the total cost, but do not affect utility. We also find diminishing returns from high information gain $s$ values ($x$-axis of Figure 2). This makes sense—as $s$ tends to infinity, the true utility is returned from a single arm pull. We also notice that if many "easy" arms (arms with very large gap scores) are allowed in higher stages, total cost rises substantially.

Although the bound defined in Theorem 1 assumes a linear function $w$, we empirically tested CACO using a submodular function $w_{\mathrm{DIV}}$. We find that the cost of running CACO using this submodular function is significantly lower than the theoretical bound. This suggests that (i) the bound for CACO can be tightened and (ii) CACO could be run with submodular functions $w$.

**BRUTAS.**   To evaluate BRUTAS, we varied $\sigma$ and $(\tilde{K}_i, T_i)$ pairs for two stages. Utility varies as expected from Theorem 2: when $\sigma$ increases, utility decreases. There is also a trade-off between $\tilde{K}_i$ and $T_i$ values. If the problem is easy, a low budget and a high $\tilde{K}_1$ value is sufficient to get high utility. If the problem is hard (high **H** value), a higher overall budget is needed, with more budget spent in the second stage. Figure 4 shows this escalating relationship between budget and utility based on problem hardness. Again we found that BRUTAS performed well when using a submodular function $w_{\mathrm{DIV}}$.

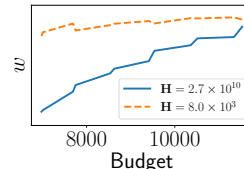

Figure 4: Effect of an increasing budget on the overall utility of a cohort. As hardness (**H**) increases, more budget is needed to produce a high quality cohort.

Finally, we compare CACO and BRUTAS to two baseline algorithms: UNIFORM and RANDOM, which uniformly and randomly respectively, pulls arms in each stage. In both algorithms, the maximization oracle is run after each stage to determine which arms should move on to the next stage. When given a budget of 2,750, BRUTAS achieves a utility of 244.0, which outperforms both the UNIFORM and RANDOM baseline utilities of 178.4 and 138.9, respectively. When CACO is run on the same problem, it finds a solution (utility of 231.0) that beats both UNIFORM and RANDOM at a roughly equivalent cost of 2,609. This qualitative behavior exists for other budgets.

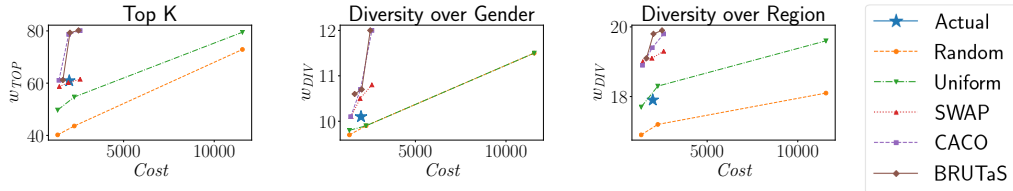

Figure 5: Utility vs Cost over four different algorithms (RANDOM, UNIFORM, SWAP, CACO, BRUTAS) and the actual admissions decisions made at the university. Both CACO and BRUTAS produce equivalent cohorts to the actual admissions process with lower cost, or produce high quality cohorts than the actual admissions process with equivalent cost.

## 5.2 Graduate Admissions Experiment

We evaluate how CACO and BRUTAS might perform in the real world by applying them to a graduate admissions dataset from one of the largest US-based graduate computer science programs. These experiments were approved by the university's Institutional Review Board and did not affect any admissions decisions for the university. Our dataset consists of three years (2014–16) worth of graduate applications. For each application we also have graduate committee review scores (normalized to between $0$ and $1$) and admission decisions.

**Experimental setup.** Using information from 2014 and 2015, we used a random forest classifier [Pedregosa *et al.*, 2011], trained in the standard way on features extracted from the applications, to predict probability of acceptance. Features included numerical information such as GPA and GRE scores, topics from running Latent Dirichlet Allocation (LDA) on faculty recommendation letters [Schmader *et al.*, 2007], and categorical information such as region of origin and undergraduate school. In the testing phase, the classifier was run on the set of applicants $A$ from 2016 to produce a probability of acceptance $P(a)$ for every applicant $a \in A$.

We mimic the university's application process of two stages: a first review stage where admissions committee members review the application packet, and a second interview stage where committee members perform a Skype interview for a select subset of applicants. The committee members follow a structured interview approach. We determined that the time taken for a Skype interview is roughly 6 times as long as a packet review, and therefore we set the cost multiplier for the second stage $j_2 = 6$. We ran over a variety of $s_2$ values, and we determined $\sigma$ by looking at the distribution of review scores from past years. When an arm $a \in A$ is pulled with information gain $s$ and cost $j$, a reward is randomly pulled from the arm's review scores (when $s_1 = 1$ and $j_1 = 1$, as in the first stage), or a reward is pulled from a Gaussian distribution with mean $P(a)$ and a standard deviation of $\frac{\sigma}{\sqrt{s}}$.

We ran simulations for BRUTAS, CACO, UNIFORM, and RANDOM. In addition we compare to an adjusted version of Schumann *et al.* [2019]'s SWAP. SWAP uses a strong pull policy to probabilistically weak or strong pull arms. In this adjusted version we use a strong pull policy of always weak pulling arms until some threshold time $t$ and strong pulling for the remainder of the algorithm. Note that this adjustment moves SWAP away from fixed confidence but not all the way to a budgeted algorithm like BRUTAS but fits into the tiered structure. For the budgeted algorithms BRUTAS, UNIFORM, and RANDOM, (as well as the pseudo-budgeted SWAP) if there are $K_i$ arms in round $i$, the budget is $K_i \cdot x_i$ where $x_i \in \mathbb{N}$. We vary $\delta$ and $\epsilon$ to control CACO's cost.

We compare the utility of the cohort selected by each of the algorithms to the utility from the cohort that was actually selected by the university. We maximize either objective $w_{\text{TOP}}$ or $w_{\text{DIV}}$ for each of the algorithms. We instantiate $w_{\text{DIV}}$, defined in Equation 5, in two ways: first, with self-reported gender, and second, with region of origin. Note that since the graduate admissions process is run entirely by humans, the committee does not explicitly maximize a particular function. Instead, the committee tries to find a good overall cohort while balancing areas of interest and general diversity.

**Results.** Figure 5 compares each algorithm to the actual admissions decision process performed by the real-world committee. In terms of utility, for both $w_{\text{TOP}}$ and $w_{\text{DIV}}$, BRUTAS and CACO achieve similar gains to the actual admissions process (higher for $w_{\text{DIV}}$ over region of origin) when using less cost/budget. When roughly the same amount of budget is used, BRUTAS and CACO are able to

provide higher predicted utility than the true accepted cohort, for both $w_{\text{TOP}}$ and $w_{\text{DIV}}$. As expected, BRUTAS and CACO outperform the baseline algorithms RANDOM, UNIFORM. The adjusted SWAP algorithm performs poorly in this restricted setting of tiered hiring. By limiting the strong pull policy of SWAP, only small incremental improvements can be made as $Cost$ is increased.

## 6  Conclusions & Discussion of Future Research

We provided a formalization of tiered structured interviewing and presented two algorithms, CACO in the PAC setting and BRUTAS in the fixed-budget setting, which select a near-optimal cohort of applicants with provable bounds. We used simulations to quantitatively explore the impact of various parameters on CACO and BRUTAS and found that behavior aligns with theory. We showed empirically that both CACO and BRUTAS work well with a submodular function that promotes diversity. Finally, on a real-world dataset from a large US-based Ph.D. program, we showed that CACO and BRUTAS identify higher quality cohorts using equivalent budgets, or comparable cohorts using lower budgets, than the status quo admissions process. Moving forward, we plan to incorporate multi-dimensional feedback (e.g., with respect to an applicant's technical, presentation, and analytical qualities) into our model; recent work due to Katz-Samuels and Scott [2018, 2019] introduces that feedback (in a single-tiered setting) as a marriage of MAB and constrained optimization, and we see this as a fruitful model to explore combining with our novel tiered system.

**Discussion.** The results support the use of BRUTAS and CACO in a practical hiring scenario. Once policymakers have determined an objective, BRUTAS and CACO could help reduce costs and produce better cohorts of employees. Yet, we note that although this experiment uses real data, it is still a simulation. The classifier is not a true predictor of utility of an applicant. Indeed, finding an estimate of utility for an applicant is a nontrivial task. Additionally, the data that we are using incorporates human bias in admission decisions, and reviewer scores [Schmader *et al.*, 2007; Angwin *et al.*, 2016]. Finally, defining an objective function on which to run CACO and BRUTAS is a difficult task. Recent advances in human value judgment aggregation [Freedman *et al.*, 2018; Noothigattu *et al.*, 2018] could find use in this decision-making framework.

## 7  Acknowledgements

Schumann and Dickerson were supported by NSF IIS RI CAREER Award #1846237. We thank Google for gift support, University of Maryland professors David Jacobs and Ramani Duraiswami for helpful input, and the anonymous reviewers for helpful comments.

## Footnotes

[1]For detailed figures see Appendix E

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
