[Supplementary Material · cut_appendix.pdf]

# A  Table of Symbols

In this section, for expository ease and reference, we aggregate all symbols used in the main paper and give a brief description of their meaning and use. We note that each symbol is also defined explicitly in the body of the paper; Table 1 is provided as a reference.

| Symbol | Summary |
|---|---|
| $n$ | Number of applicants/arms |
| $A$ | Set of all arms (e.g., the set of all applicants) |
| $a$ | An arm in $A$ (e.g., an individual applicant) |
| $K$ | Size of the required cohort |
| $\mathcal{M}_K(A)$ | Decisions class or set of possible cohorts of size $K$ |
| $u(a)$ | True utility of arm $a$ where $u(a) \in [0, 1]$ |
| $\hat{u}(a)$ | Empirical estimate of the utility of arm $a$ |
| $rad(a)$ | Uncertainty bound around the empirical estimate of the utility $\hat{u}(a)$ of arm $a$ |
| $w$ | Submodular and monotone objective function for a cohort where $w : \mathbb{R}^n \times \mathcal{M}_K(A) \to \mathbb{R}$ |
| $Oracle$ | Maximization oracle defined in Equation 1 and used by CACO |
| $COracle$ | Constrained maximization oracle used by BRUTAS |
| $M^*$ | Optimal cohort given the true utilities |
| $\Delta_a$ | The gap score of arm $a$ defined in Equation 2 |
| $\mathbf{H}$ | The hardness of a problem defined in Equation 3 |
| $j_i$ | Cost of an arm pull at stage $i$ |
| $s_i$ | Information gain of an arm pull at stage $i$ |
| $m$ | Number of pulling stages (or interview stages) |
| $K_i$ | Number of arms moving onto the next stage (stage $i + 1$) |
| $A_i$ | The active arms that move onto the next stage (stage $i + 1$) |
| $T(a)$ | Total information gain for arm $a$ |
| $\tilde{u}(a)$ | Worst case estimate of utility of arm $a$ |
| $\tilde{A}_i$ | Best cohort chosen by using the worst case estimates of utility |
| $\epsilon$ | We want to return a cohort with total utility bounded by $w(M^*) - \epsilon$ for Algorithm 1 |
| $\delta$ | The probability that we are within $\epsilon$ of the best cohort for Algorithm 1 |
| $T_i$ | Budget constraint for round $i$ |
| $T$ | Total budget |
| $T$ | Total Cost for CACO |
| $\sigma$ | Property of the $\sigma$-sub-Gaussian tailed normal distribution |
| $p$ | The arm with the greatest uncertainty in CACO |
| $\tilde{K}_i$ | Number of decisions to make in round $i$ |
| $\tilde{T}_{i,t}$ | Budget for BRUTAS in stage $i$, round $t$ |
| $M_{i,t}$ | Best cohort chosen in BRUTAS stage $i$, round $t$, using empirical utilities |
| $\tilde{M}_{i,t,a}$ | Pessimistic estimate in BRUTAS stage $i$, round $t$, for arm $a$ |
| $p_{i,t}$ | Arm which results in largest gap in BRUTAS stage $i$, round $t$ |
| $\mathbf{H}$ | Hardness for BRUTAS |
| $P(a)$ | Probability of acceptance for an arm (candidate), estimated by Random Forest Classifier |
| $q$ | Number of groups for submodular diversity function |
| $P_1, P_2, \ldots, P_q$ | The groups for submodular diversity function |

Table 1: List of symbols used in the main paper.

# B  Proofs

In this section, we provide proofs for the theoretical results presented in the main paper. Appendix B.1 gives proofs for CACO, defined as Algorithm 1 in Section 3. Appendix B.2 gives proofs for BRUTAS, defined as Algorithm 2 in Section 4.

## B.1  CACO

Theorem 1 requires lemmas from Chen *et al.* [2014]. We restate the theorem here for clarity and then proceed with the proof.

**Theorem 1.** *Given any $\delta \in (0,1)$, any $\epsilon \in (0,1)$, any decision classes $\mathcal{M}_i \subseteq 2^{[n]}$ for each stage $i \in [m]$, any linear function $w$, and any expected rewards $u \in \mathbb{R}^n$, assume that the reward distribution $\varphi_a$ for each arm $a \in [n]$ has mean $u(a)$ with a $\sigma$-sub-Gaussian tail. Let $M_i^* = \operatorname{argmax}_{M \in \mathcal{M}_i}$ denote the optimal set in stage $i \in [m]$. Set $rad_t(a) = \sigma\sqrt{2\log(\frac{4K_{i-1}\ Cost_{i,t}^3}{\delta})/T_{i,t}(a)}$ for all $t > 0$ and $a \in [n]$. Then, with probability at least $1 - \delta$, the CACO algorithm (Algorithm 1) returns the set Out where $w(\text{Out}) - w(M_m^*) < \epsilon$ and*

$$T \leq O\left(\sigma^2 \sum_{i \in [m]} \left(\frac{j_i}{s_i}\left(\sum_{a \in A_{i-1}} \min\left\{\frac{1}{\Delta_a^2}, \frac{K_i^2}{\epsilon^2}\right\}\right)\log\left(\frac{\sigma^2 j_i^4}{s_i \delta}\sum_{a \in A_{i-1}} \min\left\{\frac{1}{\Delta_a^2}, \frac{K_i^2}{\epsilon^2}\right\}\right)\right)\right).$$

*Proof.* Assume we are in some round $i$, and that we are at time $t_a$ where some arm $a$ is going to be pulled for the last time in round $i$. Set $rad_{i,t}(a) = \sigma\sqrt{\frac{2\log(4K_{i-1}\ Cost_{i,t}^3/\delta)}{T_{i,t}(a)}}$ Using Lemma 13 from Chen *et al.* [2014] we know that $rad_{i,t} \geq \max\left\{\frac{\Delta_a}{6}, \frac{\epsilon}{2K_i}\right\}$. Before arm $a$ is pulled the following must be true:

$$rad_{i,t_a} \geq \max\left\{\frac{\Delta_a}{6}, \frac{\epsilon}{2K_i}\right\} \tag{7}$$

$$rad_{i,t_a} = \sigma\sqrt{\frac{2\log(4K_{i-1}\ Cost_{i,t_a}^3/\delta)}{T_i(a) - s_i}}$$
$$\leq \sigma\sqrt{\frac{2\log(4K_{i-1}j_i^3 t_a^3/\delta)}{T_i(a) - s_i}}. \tag{8}$$

Equation 8 holds since $j_i > j_{i-1} > \cdots > j_0$. Given equations 7 and equation 8 we have,

$$\max\left\{\frac{\Delta_a}{6}, \frac{\epsilon}{2K_i}\right\} \leq \sigma\sqrt{\frac{2\log(2K_{i-1}j_i^3 t_a^3/\delta)}{T_i(a) - s_i}}$$
$$\leq \sigma\sqrt{\frac{2\log(2K_{i-1}j_i^3 T_i^3/\delta)}{T_i(a) - s_i}}$$

Solving for $T_i(a)$ we have,

$$T_i(a) \leq \sigma^2 \min\left\{\frac{72}{\Delta_a^2}, \frac{16K_i^2}{\epsilon^2}\right\}\log(4K_{i-1}j^3 T_i^3/\delta)$$
$$+ s_i \tag{9}$$

Note that

$$T_i(a) \leq T(a)$$

$$\frac{j_i}{s_i}\sum_{a \in A_{i-1}} T_i(a) = T_i. \tag{10}$$

We will show later on in the proof

$$T_i \leq 499 \frac{\sigma^2 j_i}{s_i} \left( \sum_{a \in A_{i-1}} \min\left\{ \frac{4}{\Delta_a^2}, \frac{K_i^2}{\epsilon^2} \right\} \right) \log \left( \frac{4\sigma^2 j_i^4}{s_i \delta} \sum_{a \in A_{i-1}} \min\left\{ \frac{4}{\Delta_a^2}, \frac{K_i^2}{\epsilon^2} \right\} \right)$$
$$+ 2j_i K_{i-1}. \tag{11}$$

Summing up over equation 11 we have

$$T \leq 499\sigma^2 \sum_{i \in [m]} \left( \frac{j_i}{s_i} \left( \sum_{a \in A_{i-1}} \min\left\{ \frac{4}{\Delta_a^2}, \frac{K_i^2}{\epsilon^2} \right\} \right) \log \left( \frac{4\sigma^2 j_i^4}{s_i \delta} \sum_{a \in A_{i-1}} \min\left\{ \frac{4}{\Delta_a^2}, \frac{K_i^2}{\epsilon^2} \right\} \right) \right) \tag{12}$$

which proves theorem 1.

Now we will go back to prove equation 11. If $K_{i-1} \geq \frac{1}{2} T_i$, then we see that $T_i \leq 2K_{i-1}$ and therefore equation 11 holds. Assume, then, that $K_{i-1} < \frac{1}{2} T_i$. Since $T_i > K_{i-1}$, we can write

$$T = C \frac{\sigma^2 j_i}{s_i} \left( \sum_{a \in A_{i-1}} \min\left\{ \frac{4}{\Delta_a^2}, \frac{K_i^2}{\epsilon^2} \right\} \right) \log \left( \frac{2K_{i-1}\sigma^2 j_i^4}{s_i \delta} \sum_{a \in A_{i-1}} \min\left\{ \frac{4}{\Delta_a^2}, \frac{K_i^2}{\epsilon^2} \right\} \right). \tag{13}$$

If $C < 499$ then equation 11 holds. Suppose then that $C > 499$. Using equation 10 and summing equation 12 for all active arms $a \in A_{i-1}$, we have

$$
\begin{aligned}
T_i &\leq \frac{j_i}{s_i} \left( K_{i-1} s_i + \sum_{a \in A_{i-1}} \sigma^2 \min\left\{ \frac{72}{\Delta_a^2}, \frac{16K_i^2}{\epsilon^2} \right\} \log \left( \frac{4K_{i-1} j_i^3 T_i^3}{\delta} \right) \right) \\
&\leq K_{i-1} j_i + \frac{18\sigma^2 j_i}{s_i} \left( \sum_{a \in A_{i-1}} \min\left\{ \frac{4}{\Delta_a^2}, \frac{K_i^2}{\epsilon^2} \right\} \right) \log \left( \frac{4K_{i-1} j_i^3 T_i^3}{\delta} \right) \\
&= K_{i-1} j_i + \frac{18\sigma^2 j_i}{s_i} \left( \sum_{a \in A_{i-1}} \min\left\{ \frac{4}{\Delta_a^2}, \frac{K_i^2}{\epsilon^2} \right\} \right) \log \left( \frac{4K_{i-1} j_i^3}{\delta} \right) \\
&\quad + \frac{54\sigma^2 j_i}{s_i} \left( \sum_{a \in A_{i-1}} \min\left\{ \frac{4}{\Delta_a^2}, \frac{K_i^2}{\epsilon^2} \right\} \right) \log(T) \\
&\leq K_{i-1} j_i + \frac{18\sigma^2 j_i}{s_i} \left( \sum_{a \in A_{i-1}} \min\left\{ \frac{4}{\Delta_a^2}, \frac{K_i^2}{\epsilon^2} \right\} \right) \log \left( \frac{4K_{i-1} j_i^3}{\delta} \right) \\
&\quad + \frac{54\sigma^2 j_i}{s_i} \left( \sum_{a \in A_{i-1}} \min\left\{ \frac{4}{\Delta_a^2}, \frac{K_i^2}{\epsilon^2} \right\} \right) \log \left( \frac{2C\sigma^2 j^4}{s_i \delta} \left( \sum_{a \in A_{i-1}} \min\left\{ \frac{4}{\Delta_a^2}, \frac{K_i^2}{\epsilon^2} \right\} \right) \right. \\
&\qquad \left. \log \left( \frac{4K_{i-1}\sigma^2 j_i^4}{s_i \delta} \sum_{a \in A_{i-1}} \min\left\{ \frac{4}{\Delta_a^2}, \frac{K_i^2}{\epsilon^2} \right\} \right) \right) \\
&= K_{i-1} j_i + \frac{18\sigma^2 j_i}{s_i} \left( \sum_{a \in A_{i-1}} \min\left\{ \frac{4}{\Delta_a^2}, \frac{K_i^2}{\epsilon^2} \right\} \right) \log \left( \frac{4K_{i-1} j_i^3}{\delta} \right)
\end{aligned}
\tag{14}
$$

$$+\frac{54\sigma^2 j_i}{s_i}\left(\sum_{a\in A_{i-1}}\min\left\{\frac{4}{\Delta_a^2},\frac{K_i^2}{\epsilon^2}\right\}\right)\log(2C)$$

$$+\frac{54\sigma^2 j_i}{s_i}\left(\sum_{a\in A_{i-1}}\min\left\{\frac{4}{\Delta_a^2},\frac{K_i^2}{\epsilon^2}\right\}\right)\log\left(\frac{\sigma^2 j^4}{s_i\delta}\sum_{a\in A_{i-1}}\min\left\{\frac{4}{\Delta_a^2},\frac{K_i^2}{\epsilon^2}\right\}\right)$$

$$+\frac{54\sigma^2 j_i}{s_i}\left(\sum_{a\in A_{i-1}}\min\left\{\frac{4}{\Delta_a^2},\frac{K_i^2}{\epsilon^2}\right\}\right)$$

$$\log\log\left(\frac{4K_{i-1}\sigma^2 j_i^4}{s_i\delta}\sum_{a\in A_{i-1}}\min\left\{\frac{4}{\Delta_a^2},\frac{K_i^2}{\epsilon^2}\right\}\right)$$

$$\leq\quad K_{i-1}j_i+\frac{18\sigma^2 j_i}{s_i}\left(\sum_{a\in A_{i-1}}\min\left\{\frac{4}{\Delta_a^2},\frac{K_i^2}{\epsilon^2}\right\}\right)$$

$$\log\left(\frac{4K_{i-1}\sigma^2 j^4}{s_i\delta}\sum_{a\in A_{i-1}}\min\left\{\frac{4}{\Delta_a^2},\frac{K_i^2}{\epsilon^2}\right\}\right)$$

$$+\frac{54\sigma^2 j_i}{s_i}\left(\sum_{a\in A_{i-1}}\min\left\{\frac{4}{\Delta_a^2},\frac{K_i^2}{\epsilon^2}\right\}\right)$$

$$\log(2C)\log\left(\frac{4K_{i-1}\sigma^2 j^4}{s_i\delta}\sum_{a\in A_{i-1}}\min\left\{\frac{4}{\Delta_a^2},\frac{K_i^2}{\epsilon^2}\right\}\right)$$

$$+\frac{54\sigma^2 j_i}{s_i}\left(\sum_{a\in A_{i-1}}\min\left\{\frac{4}{\Delta_a^2},\frac{K_i^2}{\epsilon^2}\right\}\right)\log\left(\frac{4K_{i-1}\sigma^2 j^4}{s_i\delta}\sum_{a\in A_{i-1}}\min\left\{\frac{4}{\Delta_a^2},\frac{K_i^2}{\epsilon^2}\right\}\right)$$

$$+\frac{54\sigma^2 j_i}{s_i}\left(\sum_{a\in A_{i-1}}\min\left\{\frac{4}{\Delta_a^2},\frac{K_i^2}{\epsilon^2}\right\}\right)\log\left(\frac{4K_{i-1}\sigma^2 j_i^4}{s_i\delta}\sum_{a\in A_{i-1}}\min\left\{\frac{4}{\Delta_a^2},\frac{K_i^2}{\epsilon^2}\right\}\right)$$

$$=\quad K_{i-1}j_i+(126+54\log(2C))\frac{\sigma^2 j_i}{s_i}\left(\sum_{a\in A_{i-1}}\min\left\{\frac{4}{\Delta_a^2},\frac{K_i^2}{\epsilon^2}\right\}\right)$$

$$\log\left(\frac{4K_{i-1}\sigma^2 j_i^4}{s_i\delta}\sum_{a\in A_{i-1}}\min\left\{\frac{4}{\Delta_a^2},\frac{K_i^2}{\epsilon^2}\right\}\right)$$

$$\leq\quad K_{i-1}j_i+C\frac{\sigma^2 j_i}{s_i}\left(\sum_{a\in A_{i-1}}\min\left\{\frac{4}{\Delta_a^2},\frac{K_i^2}{\epsilon^2}\right\}\right)$$

$$\log\left(\frac{4K_{i-1}\sigma^2 j_i^4}{s_i\delta}\sum_{a\in A_{i-1}}\min\left\{\frac{4}{\Delta_a^2},\frac{K_i^2}{\epsilon^2}\right\}\right)\tag{15}$$

$$=\quad T_i\tag{16}$$

where equation 14 follows from equation 13 and the assumption that $K_{i-1}<\frac{1}{2}T_i$; equation 15 follows since $136+54\log(2C)<C$ for all $C>499$; and 16 is due to 13. Equation 16 is a contradiction. Therefore $C\leq 499$ and we have proved equation 11.

$\square$

## B.2  BRUTAS

In order to prove Theorem 2, we first need a few lemmas.

**Lemma 1.** *Let* $\Delta_{(1)}, \ldots, \Delta_{(n)}$ *be a permutation of* $\Delta_1, \ldots \Delta_n$ *(defined in Eq. (2)) such that* $\Delta_{(1)} \leq \ldots \leq \Delta_{(n)}$. *Given a stage* $i \in [m]$, *and a phase* $t \in [\tilde{K}_i]$, *we define random event* $\tau_{i,t}$ *as follows*

$$\tau_{i,t} = \left\{ \forall i \in [n] \setminus (A_t \cup B_t) \quad |\hat{u}_{i,t}(a) - u(a)| < \frac{\Delta_{(n - \sum_{b=0}^{i-1} \tilde{K}_b - t + 1)}}{6} \right\}. \tag{17}$$

*Then, we have*

$$\tau = \Pr\left[ \bigcap_{i=1}^{m} \bigcap_{t=1}^{\tilde{K}_i} \tau_{i,t} \right] \geq 1 - n^2 \exp\left( -\frac{\sum_{b=1}^{m} s_b(\bar{T}_b - \tilde{K}_b)/(j_i \widetilde{\log}(\tilde{K}_b))}{72\sigma^2 \tilde{\mathbf{H}}} \right). \tag{18}$$

*Proof.* In round $i$ at phase $t$, arm $a$ has been pulled $\bar{T}(a)$ times. Therefore, by Hoeffding's inequality, we have

$$\Pr\left[ |\hat{u}_{i,t}(a) - u(a)| \geq \frac{\Delta_{(n - \sum_{b=0}^{i-1} \tilde{K}_b - t + 1)}}{6} \right] \leq 2 \exp\left( -\frac{\bar{T}_{i,t}(a)\Delta_{(n - \sum_{b=0}^{i-1} \tilde{K}_b - t + 1)}^2}{72\sigma^2} \right) \tag{19}$$

By using the definition of $\tilde{T}_{i,t}$, the quantity $\tilde{T}_{i,t}\Delta_{(n - \sum_{b=1}^{\tilde{K}_{i-1}} \tilde{K}_b - t + 1)}^2$ on the right-hand side of Eq. 19 can be further bounded by

$$\bar{T}_{i,t}\Delta_{(n - \sum_{b=1}^{\tilde{K}_{i-1}} \tilde{K}_b - t + 1)}$$

$$= (s_i \tilde{T}_{i,t} + \sum_{b=1}^{i-1} s_b \tilde{T}_{b, \tilde{K}_b + 1}) \Delta_{(n - \sum_{b=1}^{\tilde{K}_{i-1}} \tilde{K}_b - t + 1)}^2$$

$$\geq \left( \frac{s_i(\bar{T}_{i,t} - \tilde{K}_i)}{j_i \widetilde{\log}(\tilde{K}_i)(\tilde{K}_i - t + 1)} + \sum_{b=0}^{i-1} \frac{s_b(\bar{T}_{b, \tilde{K}_b + 1} - \tilde{K}_b)}{j_b \widetilde{\log}(\tilde{K}_b)(\tilde{K}_b - \tilde{K}_b + 2)} \right) \Delta_{(n - \sum_{b=1}^{i-1} \tilde{K}_b - t + 1)}^2$$

$$\geq \sum_{b=1}^{i} \frac{s_b(\bar{T}_b - \tilde{K}_b)}{j_b \widetilde{\log}(\tilde{K}_b)\tilde{\mathbf{H}}},$$

where the last inequality follows from the definition of $\tilde{\mathbf{H}} = \max_{i \in [n]} i\Delta_{(i)}^{-2}$. By plugging the last inequality into Eq. 19, we have

$$\Pr\left[ |\hat{u}_{i,t}(a) - u(a)| \geq \frac{\Delta_{(n - \sum_{b=0}^{i-1} \tilde{K}_b - t + 1)}}{6} \right] \leq 2 \exp\left( -\frac{\sum_{b=1}^{i} \left( s_b(\bar{T}_b - \tilde{K}_b)/(j_b \widetilde{\log}(\tilde{K}_b)) \right)}{72\sigma^2 \tilde{\mathbf{H}}} \right) \tag{20}$$

Now, using Eq. 20 and a union bound for all $i \in [m]$, all $t \in [\tilde{K}_i]$, and all $a \in [n] \setminus (A_{t,i} \cup B_{t,i})$, we have

$$\Pr\left[ \bigcap_{i=1}^{m} \bigcap_{t=1}^{\tilde{K}_i} \tau_{i,t} \right]$$

$$\geq 1 - 2 \sum_{i=1}^{m} \sum_{t=1}^{\tilde{K}_i} (n - \sum_{b=0}^{i-1} \tilde{K}_b - t + 1) \exp\left( -\frac{\sum_{b=1}^{i} \left( s_b(\bar{T}_b - \tilde{K}_b)/(j_b \widetilde{\log}(\tilde{K}_b)) \right)}{72\sigma^2 \tilde{\mathbf{H}}} \right)$$

$$\geq 1 - n^2 \exp\left( -\frac{\sum_{b=1}^{m} s_b(T_b - K_b)/(j_i \widetilde{\log}(K_i))}{72\sigma^2 \mathbf{H}} \right).$$

$\square$

**Lemma 2.** *Fix a stage $i \in [m]$, and a phase $t \in [\tilde{K}_i]$, suppose that random event $\tau_{i,t}$ occurs. For any vector $\mathbf{a} \in \mathbb{R}^n$, suppose that $\mathrm{supp}(\mathbf{a}) \cap (A_{i,t} \cup B_{i,t} = \varnothing$, where $\mathrm{supp}(\mathbf{a}) \triangleq \{i|a(i) \neq 0\}$ is the support of vector $\mathbf{a}$. Then, we have*

$$|\langle \tilde{\mathbf{u}}_{i,t}, \mathbf{a} \rangle - \langle \mathbf{u}_{i,t}, \mathbf{a} \rangle| < \frac{\Delta_{(n-\sum_{b=0}^{i-1}\tilde{K}_i-t+1)}}{6}\|\mathbf{a}\|_1$$

*Proof.* Suppose that $\tau_{i,t}$ occurs. Then, we have

$$|\langle \tilde{\mathbf{u}}_{i,t}, \mathbf{a} \rangle - \langle \mathbf{u}_{i,t}, \mathbf{a} \rangle| \tag{21}$$
$$= |\langle \tilde{\mathbf{u}}_{i,t} - \mathbf{w}, \mathbf{a}|$$
$$= \left| \sum_{b=1}^{n} (\tilde{u}_{t,i}(b) - u(b))\, a(b) \right|$$
$$\leq \left| \sum_{b \in [n]\setminus(A_{i,t}\cup B_{i,t})} (\tilde{u}_{i,t}(b) - u(b))\, a(b) \right| \tag{22}$$
$$\leq \sum_{b \in [n]\setminus(A_{i,t}\cup B_{i,t})} |(\tilde{u}_{i,t}(b) - u(b))\, a(b)|$$
$$\leq \sum_{b \in [n]\setminus(A_{i,t}\cup B_{i,t})} |\tilde{u}_{i,t}(b) - u(i)||a(i)|$$
$$< \frac{\Delta_{(n-\sum_{b=0}^{i-1}\tilde{K}_i-t+1)}}{6} \sum_{b \in [n]\setminus(A_{i,t}\cup B_{i,t})} |a(b)| \tag{23}$$
$$= \frac{\Delta_{(n-\sum_{b=0}^{i-1}\tilde{K}_i-t+1)}}{6}\|\mathbf{a}\|_1 \tag{24}$$

where Eq. 22 follows from the assumption that $\mathbf{a}$ is supported on $[n] \setminus (A_{i,t} \cup B_{i,t})$; Eq. 23 follows from the definition of $\tau_{i,t}$ (Eq. 17). $\square$

**Lemma 3.** *Fix a stage $i \in [m]$, and a phase $t \in [\tilde{K}_i]$. Suppose that $A_{i,t} \subseteq M_*$ and $B_{i,t} \cap M_* = \varnothing$. Let $M$ be a set such that $A_{i,t} \subseteq M$ and $B_{i,t} \cap M = \varnothing$. Let $a$ and $b$ be two sets satisfying $a \subseteq M \setminus M_*$, $b \subseteq M_* \setminus M$, and $a \cap b = \varnothing$. Then, we have*

$$A_{i,t} \subseteq (M \setminus a \cup b)$$

*and*

$$B_{i,t} \cap (M \setminus a \cup b) = \varnothing$$

*and*

$$(a \cup b) \cap (A_{i,t} \cup B_{i,t}) = \varnothing.$$

Lemma 3 is due to Chen *et al.* [2014].

**Lemma 4.** *Fix any stage $i \in [m]$, and any phase $t \in [\tilde{K}_i]$ such that $\sum_{b=0}^{i-1}\tilde{K}_i + t > 0$. Suppose that event $\tau_{i,t}$ occurs. Also assume that $A_{i,t} \subseteq M_*$ and $B_{i,t} \cap M_* = \varnothing$. Let $a \in [n] \setminus (A_{i,t} \cup B_{i,t})$ be an active arm such that $\Delta_{(n-\sum_{b=0}^{i-1}K_i-t+1)} \leq \Delta_a$. Them, we have*

$$\tilde{u}_{i,t}(M_{i,t}) - \tilde{u}_{i,t}(\tilde{M}_{i,t,a}) > \frac{2}{3}\Delta_{(n-\sum_{b=0}^{i-1}\tilde{K}_i-t+1)}$$

Lemma 4 is due to Chen *et al.* [2014].

**Lemma 5.** *Fix any stage $i \in [m]$, and any phade $t \in [K_i]$ such that $\sum_{b=0}^{i-1}K_i + t > 0$. Suppose that event $\tau_{i,t}$ occurs. Also assume that $A_{i,t} \subseteq M_*$ and $B_{i,t} \cap M_* = \varnothing$. Suppose an active arm $a \in [n] \setminus (A_{i,t} \cap B_{i,t})$ satisfies that $a \in (M_* \cap \neg M_{i,t}) \cup (\neq M_* \cap M_{i,t})$. Then, we have*

$$\tilde{u}_{i,t}(M_{i,t}) - \tilde{u}_{i,t}(\tilde{M}_{i,t,a}) \leq \frac{1}{3}\Delta_{(n-\sum_{b=0}^{i-1}K_i-t+1)}$$

Lemma 5 is due to Chen *et al.* [2014].

Now we can prove Theorem 2, restated below for clarity.

**Theorem 2.** *Given any $\bar{T}_i$s such that $\sum_{i\in[m]}\bar{T}_i = \bar{T} > n$, any decision class $\mathcal{M}_K \subseteq 2^{[n]}$, any linear function $w$, and any true expected rewards $\mathbf{u} \in \mathbb{R}^n$, assume that reward distribution $\varphi_a$ for each arm $a \in [n]$ has mean $u(a)$ with a $\sigma$-sub-Gaussian tail. Let $\Delta_{(1)}, \ldots, \Delta_{(n)}$ be a permutation of $\Delta_1, \ldots, \Delta_n$ (defined in Eq. 2) such that $\Delta_{(1)} \leq \ldots \leq \Delta_{(n)}$. Define $\tilde{\mathbf{H}} \triangleq \max_{i\in[n]} i\Delta_{(i)}^{-2}$. Then, Algorithm 2 uses at most $\bar{T}_i$ samples per stage $i \in [m]$ and outputs a solution $\mathtt{Out} \in \mathcal{M}_K \cup \{\perp\}$ such that*

$$\Pr[\mathtt{Out} \neq M_*] \leq n^2 \exp\left(-\frac{\sum_{b=1}^{m} s_b(\bar{T}_b - \tilde{K}_b)/(j_b \widetilde{\log}(\tilde{K}_b))}{72\sigma^2 \tilde{\mathbf{H}}}\right) \tag{25}$$

*where $\widetilde{\log}(n) \triangleq \sum_{i=1}^{n} i^{-1}$, and $M_* = \operatorname{argmax}_{M \in \mathcal{M}_K} w(M)$.*

*Proof.* First we show that the algorithm takes at most $\bar{T}_i$ samples in every stage $i \in [m]$. It is easy to see that exactly one arm is pulled for $\tilde{T}_{i,1}$ times in stage $i$, one arm is pulled for $\tilde{T}_{i,2}$ times in stage $i$, ..., and one arm is pulled for $\tilde{T}_{i,\tilde{K}_i - 1}$ times in stage $i$. Therefore, the total number of samples used by the algorithm in stage $i \in [m]$ is bounded by

$$
\begin{aligned}
j_i \sum_{t=1}^{\tilde{K}_i} \tilde{T}_{i,t} &\leq j_i \sum_{t=1}^{\tilde{K}_i} \left( \frac{\bar{T}_i - \tilde{K}_i}{\widetilde{\log}(\tilde{K}_i)j_i(\tilde{K}_i - t + 1)} + 1 \right) \\
&= \frac{(\bar{T}_i - \tilde{K}_i)j_i}{\widetilde{\log}(\tilde{K}_i)j_i} \widetilde{\log}(\tilde{K}_i) + \tilde{K}_i \\
&= \bar{T}_i.
\end{aligned}
$$

By Lemma 1, we know that the event $\tau$ occurs with probability at least $1 - n^2 \exp\left(-\frac{\sum_{b=1}^{m} s_b(\bar{T}_b - \tilde{K}_b)/(j_i \widetilde{\log}(\tilde{K}_i))}{72\sigma^2 \tilde{\mathbf{H}}}\right)$. Therefore, we only need to prove that, under event $\tau$, the algorithm outputs $M_*$. Assume that the event $\tau$ occurs in the rest of the proof.

We will use induction. Fix a stage $i \in [m]$ and phase $t \in [\tilde{K}_i]$. Suppose that the algorithm does not make any error before stage $i$ and phase $t$, i.e. $A_{i,t} \subseteq M_*$ and $B_{i,t} \cap M_* = \varnothing$. We will show that the algorithm does not err at stage $i$, phase $t$.

At the beginning of phase $t$ in stage $i$ there are exactly $\sum_{b=0}^{i-1} \tilde{K}_i + t - 1$ inactive arms $|A_{i,t} \cup B_{i,t}| = \sum_{b=0}^{i-1} \tilde{K}_i + t - 1$. Therefore there must exist an active arm $e_{i,t} \in [n] \setminus (A_{i,t} \cup B_{i,t})$ such that $\Delta_{e_{i,t}} \geq \Delta_{(n-\sum_{b=0}^{i-1} \tilde{K}_i - t + 1)}$. Hence, by Lemma 4, we have

$$\tilde{w}_{i,t}(M_{i,t}) - \tilde{w}_{i,t}(M_{i,t,e_{i,t}}) \geq \frac{2}{3}\Delta_{(n-\sum_{b=0}^{i-1} \tilde{K}_i - t + 1)}. \tag{26}$$

Notice that the algorithm makes an error in phase $t$ in stage $i$ if and only if it accepts an arm $p_{i,t} \notin M_*$ or rejects an arm $p_{i,t} \in M_*$. On the other hand, arm $p_{i,t}$ is accepted when $p_{i,t} \in M_{i,t}$ and is rejected when $p_{i,t} \notin M_{i,t}$. Therefore, the algorithm makes an error in phase $t$ in stage $i$ if and only if $p_t \in (M_* \cap \neg M_{i,t}) \cup (\neg M_* \cap M_{i,t})$.

Suppose that $p_t \in (M_* \cap \neg M_{i,t}) \cup (\neg M_* \cap M_{i,t})$. Using Lemma 5, we see that

$$\tilde{w}_{i,t}(M_{i,t}) - \tilde{w}_{i,t}(\tilde{M}_{i,t,p_{i,t}}) \leq \frac{1}{3}\Delta_{(n-\sum_{b=0}^{i-1} \tilde{K}_i - t + 1)}. \tag{27}$$

By combining Eq. 26 and Eq. 27, we see that

$$\tilde{w}_{i,t}(M_{i,t}) - \tilde{w}_{i,t}(\tilde{M}_{i,t,p_{i,t}}) \tag{28}$$

$$\leq \frac{1}{3}\Delta_{(n-\sum_{b=0}^{i-1} \tilde{K}_i - t + 1)} \tag{29}$$

$$< \frac{2}{3}\Delta_{(n-\sum_{b=0}^{i-1} \tilde{K}_i - t + 1)} \tag{30}$$

$$\leq \tilde{w}_{i,t}(M_{i,t}) - \tilde{w}_{i,t}(\tilde{M}_{i,t,e_{i,t}}) \tag{31}$$

(a) $\delta$: 0.075, $\epsilon$: 0.075        (b) $\delta$: 0.3, $\epsilon$: 0.075        (c) $\delta$: 0.075, $\epsilon$: 0.3

Figure 6: Hardness ($\mathbf{H}$) vs theoretical cost ($T$) as user-specified parameters to the CACO algorithm.

However, Eq. 28 is contradictory to the definition of $p_{i,t} \triangleq \mathrm{argmax}_{e \in [n] \setminus (A_{i,t} \cup B_{i,t})} \tilde{w}_{i,t}(M_{i,t}) - \tilde{w}_{i,t}(M_{i,t,e})$. This proves that $p_t \notin (M_* \cap \neg M_{i,t}) \cup (\neg M_* \cap M_{i,t})$. This means that the algorithm does not err at phase $t$ in stage $i$, or equivalently $A_{i,t+1} \subseteq M_*$ and $B_{i,t+1} \cap M_* = \varnothing$.

Hence we have $A_{m,\tilde{K}_{m+1}} \subseteq M_*$ and $B_{m,\tilde{K}_{m+1}} \subseteq \neg M_*$ in the final phase of the final stage. Notice that $|A_{m,\tilde{K}_{m+1}}| + |B_{m,\tilde{K}_{m+1}}| = n$ and $A_{m,\tilde{K}_{m+1}} \cap B_{m,\tilde{K}_{m+1}} = \varnothing$. This means that $A_{m,\tilde{K}_{m+1}} = M_*$ and $B_{m,\tilde{K}_{m+1}} = \neg M_*$. Therefore the algorithm outputs $\mathtt{Out} = A_{m,\tilde{K}_{m+1}} = M_*$ after phase $\tilde{K}_m$ in stage $m$. $\qquad\qquad\qquad\qquad\qquad\qquad\qquad\qquad\qquad\qquad\qquad\qquad\qquad\qquad\qquad\square$

## C  Visualization of CACO bound

Figure 6 shows how the theoretical bound defined in Theorem 1 changes as parameters change vs. Hardness $\mathbf{H}$ defined in Equation 3.

## D  Experimental Setup

The machines used for the experiments had 32GB RAM, 8 Intel SandyBridge CPU cores, and were initialized with Red Hat Enterprise Linux 7.3. A single run of SWAP over the graduate admissions data takes about 1 minute depending on the parameters.

| Parameter | Range |
|---|---|
| $\delta$ | 0.3,0.2,0.1,0.075,0.05 |
| $\epsilon$ | 0.3,0.2,0.1,0.075,0.05 |
| $\sigma$ | 0.1,0.2 |
| $j$ | 6 |
| $s$ | $7,\ldots,20$ |

Table 2: Parameters for graduate admissions experiments

## E  Additional Experimental Results

In this section, we present additional experimental results for CACO and BRUTAS. Table 3 supports the Gaussian simulation experiments of Section 5.1, spcefically, the comparison of CACO and BRUTAS to two baseline pulling strategies.

Table 4 also supports the Gaussian simulation experiments from Section 5.1. Here, we vary $\delta$ instead of $\epsilon$, as was done in Figure 2 of the main paper. As expected, when $\delta$ increases, the cost decreases. However, the magnitude of the effect is smaller than the effect from decreasing $\epsilon$ or varying $K_1$. This is also expected, as discussed in the final paragraphs of Section 3, and shown in Figure 1.

Figure 7 shows that, as the standard deviation $\sigma$ of the Gaussian distribution from which rewards are drawn increases, so too does the total cost of running CACO. The qualitative behavior shown in, e.g., Figure 2 of the main paper remains: as information gain $s$ increases, overall cost decreases; as $s$

| Algorithm | Cost | Utility |
|-----------|------|---------|
| Random | 2750 | 138.9 (5.1) |
| Uniform | 2750 | 178.4 (0.2) |
| CACO | 2609 | 231.0 (0.1) |
| BRUTAS | 2750 | 244.0 (0.1) |

Table 3: Comparing CACO and BRUTAS to the baseline of Uniform and Random

| $\delta$ | Cost | | | |
|---|---|---|---|---|
| | $K_1 = 10$ | $K_1 = 13$ | $K_1 = 18$ | $K_1 = 29$ |
| 0.050 | 552.475 | 605.250 | 839.525 | 1062.725 |
| 0.075 | 542.425 | 582.675 | 827.025 | 1040.700 |
| 0.100 | 537.175 | 587.900 | 820.575 | 1078.975 |
| 0.200 | 503.650 | 568.300 | 801.525 | 1012.550 |

Table 4: Cost for CACO over various $\delta$, for $\epsilon = 0.05, \sigma = 0.20, s_2 = 7$

increases substantially, we see a saturation effect; and, as final cohort size $K$ increases, overall cost increses.

Figure 8 shows the behavior of CACO for different arm initializations, representing different utilities and groupings. We chose 4 representative initializations. For most initializations, when $K_1 = 10$, higher values of $s_2$ do not result in gains. This is because with $K_1 = 10$ and $K = 7$, there are only 3 decisions to make on which arms to cut and the information gain from the initial pull of all arms in stage 2 grants enough information, thus no additional pulls need to be made and cost is uniform across $s_2$. However, if the problem of selecting from the short list is hard enough, additional resources must be spent to narrow the decisions down, as in the top left graph, where total costs decrease as $s_2$ increases for $K_1 = 10$ because additional pulls need to be made after the initial pulls of remaining arms in stage 2. This reflects real life well: usually, the short list can be cut down with one additional round of (more informative) interviews. However, in rare situations, some candidates are so close to each other that additional assessments need to be made about them. Another interesting result is that $K_1 = 10$ is not always the most cost effective option. If many of the initial candidates are close

Figure 7: Comparison of $Cost$ over information gain ($s$) as $\sigma$ increases for CACO. Here, $\delta = 0.05$ and $\epsilon = 0.05$.

Figure 8: Comparison of $Cost$ over information gain ($s$) for different sets of arms for CACO. Here, $\delta = 0.075$, $\epsilon = 0.05$, $\sigma = 0.2$.

together in utility, it will be hard to narrow it down to a final 10 based on resume review alone: more candidates should be allowed to move onto the next round which has higher information gain. This can be seen in the bottom right graph.

## F   Limitations

This experiment uses real data but is still a simulation. The classifier is not a true predictor of utility of an applicant. Indeed, finding an estimate of utility for an applicant is a nontrivial task. Additionally, the data that we are using incorporates human bias in admission decisions, and reviewer scores. This means that the classifier—and therefore the algorithms—may produce a biased cohort. Training a human committee or using quantitative methods to (attempt to) mitigate the impact of human bias in review scoring is important future work. Similarly, CACO and BRUTAS require an objective function to run; recent advances in human value judgment aggregation [Freedman *et al.*, 2018; Noothigattu *et al.*, 2018] could find use in this decision-making framework. Additionally, although we were able to empirically show that both CACO and BRUTAS perform well using a submodular function $w_{\text{DIV}}$, there are no theoretical guarantees for submodular functions.

## G   Structured Interviews for Graduate Admissions

The goal of the interview is to help judge whether the applicant should be granted admission. The interviewer asks questions to provide insight into the applicant's academic capabilities, research experience, perseverance, communication skills, and leadership abilities, among others.

Some example questions include:

- Describe a time when you have faced a difficult academic challenge or hurdle that you successfully navigated. What was the challenge and how did you handle it?
- What research experience have you had? What problem did you work on? What was most challenging? What did you learn most from the experience?
- Have you had any experiences where you were playing a leadership or mentoring role for others?
- What are your goals for graduate school? What do you want to do when you graduate?

| | Experiment | Cost | $w_{\text{TOP}}$ | $w_{\text{DIV}}$ over Gender | $w_{\text{DIV}}$ over Region of Origin |
|---|---|---|---|---|---|
| Actual | – | ~2,000 | 60.9 | 10.1 | 17.9 |
| RANDOM | lower | 1,359 | 40.2 (0.3) | 9.7 (0.2) | 16.9 (0.3) |
| | ~equivalent | 2,277 | 43.6 (0.5) | 9.9 (0.1) | 17.2 (0.2) |
| | higher | 11,556 | 72.9 (4.9) | 11.5 (0.1) | 18.1 (3.5) |
| UNIFORM | lower | 1,359 | 49.7 (0.3) | 9.8 (0.1) | 17.7 (0.1) |
| | ~equivalent | 2,277 | 54.7 (0.3) | 9.9 (0.2) | 18.3 (0.4) |
| | higher | 11,556 | 79.5 (3.2) | 11.9 (0.3) | 19.6 (0.6) |
| SWAP | lower | 1,400 1,500 | 58.7 (0.5) | 10.1 (0.1) | 19.0 (0.1) |
| | ~equivalent | 1,900–2,000 | 60.2 (0.4) | 10.5 (0.1) | 19.1 (0.1) |
| | higher | 2,500–2,700 | 61.5 (0.5) | 10.8 (0.2) | 19.3 (0.1) |
| CACO | lower | 1,400–1,460 | 61.1 (0.1) | 10.1 (0.2) | 18.9 (0.1) |
| | ~equivalent | 1,950–1,990 | 78.7 (0.2) | 10.7 (0.1) | 19.4 (0.2) |
| | higher | 2,500–2,700 | 80.1 (0.4) | 12.0 (0.3) | 19.8 (0.3) |
| BRUTAS | lower | 1,649 | 61.2 (0.2) | 10.6 (0.1) | 19.1 (0.2) |
| | ~equivalent | 2,038 | 79.3 (0.3) | 10.7 (0.1) | 19.8 (0.3) |
| | higher | 2,510 | 80.2 (0.3) | 12.0 (0.2) | 19.9 (0.2) |

Table 5: Utility vs Cost over five different algorithms (RANDOM, UNIFORM, SWAP, CACO, BRUTAS) and the actual admissions decisions made at our university. (Since CACO is a probabilistic method, the cost is given over a range of values.) For each of the algorithms, we give results assuming a cost/budget *lower*, roughly *equivalent*, and *higher* than that used by the real admissions committee. Both CACO and BRUTAS produce equivalent cohorts to the actual admissions process with lower cost, or produce high quality cohorts than the actual admissions process with equivalent cost. Our extension of SWAP to this multi-tiered setting also performs well relative to RANDOM and UNIFORM, but performs worse than both CACO and BRUTAS across the board.

- What concerns do you have about the program? What will your biggest challenge be? Is there anything else we should discuss?

The interviewer fills out an answer and score sheet during the interview. Each interviewer follows the same questions and is provided with the same answer and score sheet. This allows for consistency across interviews.