[Reviews · NeurIPS 2019]

Reviewer 1



This paper explores the problem of tiered interviewing and formulates the problem as a multi-stage combinatorial pure exploration bandit framework. The authors propose two algorithms, CACO and BRUTAS, that extends the algorithms by Chen et al., CLUCB and CSAR, to multiple stages. The authors provide theoretical bounds for the proposed algorithms and conduct simulations based on real data to evaluate the algorithms. On the positive side, I think the research question is interesting, and the proposed approach makes sense on a high level. However, there seems to be a gap between the bandit formulation and the motivating application. I also have some concerns on the technical content (mainly due to the exposition). I would lean towards rejecting the paper in its current form. == Motivation == I find the motivating problem, tiered-interviewing, really interesting. However, I have a hard time connecting the proposed bandit formulation to this problem. In particular, in the bandit formulation, at each stage, the learner can choose to pull each arm different number of times based on the responses of previous pulls. What does this arm pull mean in the interviewing process? As in typical interviewing scenarios, it seems the firm only interacts with each candidate once in each stage (with each stage being resume screening, phone interview, or on-site interview)? I think it would help to provide more discussion on how the formulation connects to the motivating example. == Technical content == While the paper is easy to follow on a high level, I have a couple of confusions and concerns on the technical content. In particular, - The definition of information gain s is only mentioned in the description of related work on page 3. Since it’s one of the more important notations, it would help to clearly define it in the setting. - What are the intuitions or interpretations on why there is a Cost^3 factor when calculating the confidence radius in Line 9 of Algorithm 1? Is the learner aiming to minimize cost or need to satisfy some budget constraint? Without explicitly stating the objective, it’s hard to interpret this choice of confidence radius. - What is T in Theorem 1? It’s not defined in the main text. Looking at Equation 11 to 13 in the appendix, it seems T is the cost divided by the information gain (this partially answers my question above, since Theorem 1 implies that the objective is to bound T). However, if this is the definition, it is not consistent with the notion T in algorithm 2, in which T is the total budget (not divided by information gain). Moreover, since T(a) is also used as the information gain for arm a, this set of notations is a bit confusing. - Algorithm 2 doesn’t seem to satisfy the budget constraint. If we set both \tilt{K}_1 and j_1 to 1, in the first stage, since no arms are in A or B yet, there will be n * (T_1 -1) arms pulls in stage 1 according to Line 6 and Line 8 of Algorithm 2, while the budget is only T_1. I might be mistaken, but it would help to formally demonstrate why Algorithm 2 satisfies the budget constraint since it’s not clear from the current description. - It seems in Algorithm 2, the learner not only needs to decide how much budget T_i to allocate to each stage, it also needs to determine \tilt{K}_i, i.e., how many arms to accept/reject. While the authors provide a brief discussion after Theorem 2, it’s not clear to me how the learner should choose these values simultaneously. - All the K in Theorem 2 should be \tilt{K}? == Experiments == The experiments are conducted under strong assumptions, which make the results less insightful. In particular, to compare the proposed algorithms with the committee decision, it is assumed the learned P(a) represent the true acceptance rate of each candidate. Moreover, it is not clear to me how to interpret the comparison with the decision by the committee, since the objective of the committee might not be the same as the learning algorithms.

Reviewer 2



The multi-round interview scenario is interesting, and has potentially applications beyond hiring. The algorithmic and theoretical extension from [Chen 2014] is non-trivial, the two algorithms are sound, with clear intuitions and computationally feasible. The two settings are analogous to those in [Chen 2014], but with intuitions updated for multi-round setting. The writing is generally clear. ------ update after author feedback ----------- I read other reviews and author feedback, and feel that the feedback helps provide perspective and have addressed concerns raise in the reviews. Assuming the explanations and additional info in the feedback makes to the final version, I maintain an accept rating.

Reviewer 3



I have read the other reviews and the author feedback. I'm curious to know the results of the comparison to the Greedy baseline that the authors promise in their response, it is hard to make a judgement without that comparison. I maintain a borderline accept rating, if the discussion points from the rebuttal make it to the final version. The discussion about how a policy-make might set K_i (size of the filter at each tier) in the response is very under-developed -- perhaps it is better to highlight and discuss this more as future work. Overall, the response addressed some of my concerns (re: practicality for the admissions setting, clarification of notation and correctness). ----- Clarity: Good. The paper was easy to follow, and the list of symbols in the appendix was a useful reference to quickly recap what the various symbols meant. The experiments and figures need to be explained more (see detailed comments). Quality: Average. There are a few natural approaches for the problem setting that could be discussed. Greedy approaches should be a baseline in the experiments (with optimistic initialization, they often perform better than Uniform or Random policies in bandit settings). Significance: Average. The problem setting is reasonably well-motivated but can be better. The algorithms that are developed do not yet seem practical for the motivating application. The results can be more significant if there is a better characterization of the gap (in terms of cost, utility, etc.) between direct Top-K selection vs. Tiered Top-K selection. This might even help illuminate what K to pick in each tier -- there is qualitative discussion about this after the synthetic experiments, the paper can be substantially stronger with a more rigorous analysis of this phenomenon. Nitpick: I did not try running the code, but there are comments (see line 358) that say "Problem here". Does it implement the pseudo-code from the paper, or is there a problem there? Does it work?! Line 176: The theorem applies for linear w. In Line 13 of Alg1, I understand w to be like w_div of Eqn5 which is clearly not modular/linear but submodular. Please clarify the theorem statement and foreshadow that w will not be linear and the theorem won't technically apply, rather than waiting till the discussion section in the end to discuss this. Line280: The approach assumes a maximization oracle. However, we can only approximately maximize submodular functions efficiently (e.g. 1/e-approximation using a greedy strategy). Can the theorem account for sub-optimality of the oracle? For instance, Radlinski et al 2008 does such an analysis for submodular ranking. Figure1: Please clarify whether Hardness is exactly the term inside the O() bound of Theorem1. Please explain how Figure 1 was generated. The paper can benefit from a discussion on why simpler approaches than CACO are doomed to fail. Either a formal lower bound showing the hardness of this problem of subset selection. Or, at least an instructive example why greedy elimination in early stages can yield unrecoverable errors in later stages. That can then motivate the notion of pessimistic estimates used in Line10-11 of Alg1. Line285: Hardness really needs to be defined more clearly! How does greedy subset selection perform? A naive baseline that does not do pessimistic corrections of the empirical estimates but simply picks items according to their empirical mean. This seems like a marginally better baseline than UNIFORM or RANDOM. Nitpick: The discussion in lines 358-360 should be highlighted earlier. The last statement in the abstract setup some unrealistic expectations. The expt in Section5.2 is better described as a semi-synthetic simulation (rather than how the algorithms might perform in the real world). Since both algorithms need multiple pulls of active arms in each stage (please clarify in Section5.2 or the appendix what is the maximum number of times an arm gets pulled by CACO and BRUTAS), these algorithms do not yet seem practical for the tiered interview setting.

[Author Response · NeurIPS 2019]

We thank all the reviewers (**R1**, **R2**, and **R4**) for their thoughtful comments, and respond to questions below.

**Hiring as a motivating application & multiple arm pulls per stage (R1, R4):** We were initially motivated by the graduate admissions system run at our university. Here, at every stage, it is possible for *multiple* independent reviewers to look at an applicant. Indeed, our admissions committee strives to hit at least two written reviews per application package, before potentially considering one or more Skype/Hangouts calls with a potential applicant. (In our data, for instance, some applicants received up to 6 independent reviews per stage.)

While motivated by academic admissions, we believe our model is of broad interest to industry as well. For example, in tech, it is common to allocate more (or fewer) 30-minute one-on-one interviews on a visit day, and/or multiple pre-visit programming screening teleconference calls. Similarly, in management consulting (cf. the McKinsey citation in our submission), it is common to repeatedly give independent "case study" interviews to borderline candidates.

**Information gain (R1):** Information gain $s$ relates to the confidence of accept/reject from an interview vs review. As stages get more expensive, the estimates of utility become more precise, i.e. the estimate comes with a distribution with a lower variance. In practice, a resume review may make a candidate seem much stronger than they are, or a badly written resume could severely underestimate their abilities. However, in-person interviews give better estimates. A strong arm pull with information gain $s$ is equivalent to $s$ weak pulls because it is equivalent to pulling from a distribution with a $\sigma/\sqrt{s}$ sub-Gaussian tail, i.e. it is equivalent to getting a (probably) closer estimate.

**Hardness (R1, R2, R4):** Hardness is defined in Eq. 3 as the sum of inverse squared gaps. We will be sure to point to Eq. 3 every time hardness is discussed; the example to be added in response to **R2** will help with intuition as well.

**Figure 1 (R4):** Fig. 1 is plotted using fixed values for $K_1$, $\delta$, and $\epsilon$. Hardness is defined in Eq. 3 (see above). We did, however, run experiments to see how the figure shifts when increasing/decreasing each of these values (to quantify the intuition given by Thm. 1). In order to save space we described the effects of the variation on the figure. In the camera ready version we can include a full page of varied graphs in the appendix, the data for which already exists.

**Exposition clarification for $T$ and budget (R1):** The definition of $T$ changes based on the section of the paper; we will change this. In Thm. 1 and Alg. 1 the reviewer is correct to assume that $T$ is the cost divided by the information gain. In Thm. 2 and Alg. 2, as presently written, $T$ becomes the budget. We will change $T$ to $\tilde{T}$ in Sec. 4 for clarity.

**Budget and $K_i$ (R1):** The reviewer is correct in suggesting that choosing budget and $K_i$ simultaneously is difficult. The policy maker should look at past decisions to estimate gap scores (Eq. 2) and hardness (Eq. 3). There is a clear trade-off between information gain and cost. If the policy maker assumes (based on past data) that the gap scores will be high (it is easy to differentiate between applicants) then the lower stages should have a high $K_i$, and a budget to match the relevant cost $j_i$. If the gap scores are all low (it is hard to differentiate between applicants) then more decisions should be made in the higher, more expensive stages. By looking at the ratio of small gap scores to high gap scores, or by bucketing gap scores, a policy maker will be able to set each $K_i$.

**Oracle and suboptimality (R4):** The algorithm assumes that the oracle is correct (which is a very light assumption for the theory section, which assumes a linear objective, but less trivial for general monotone submodular objectives like the diversity-promoting objective used in some of the experiments). However, we find that even with this assumption the algorithm performs well empirically under the submodular function given.

**Correctness of Alg. 2 (R1):** Thanks for pointing this out! It is a typo. $\tilde{K}_i$ should be $n - \sum_{a=0}^{i-1} \tilde{K}_a$, i.e., the number of arms remaining. In our code we implemented $\tilde{K}_i$ in that correct way. (See grid test on line 482 in `cut_neurips.py`).

**Cost (R1):** The confidence radius was chosen due to a Hoeffding bound. See Appendix B1 for further information.

**BRUTaS and CACO differences (R2):** The CACO algorithm is a fixed confidence algorithm, meaning that each stage ends after some (user-supplied) confidence interval is satisfied. When a stage ends, only $K_i$ *arms* move on to the next stage. The BRUTaS algorithm is a fixed budget algorithm where each stage uses a certain amount of *budget*. All arms are pulled $\tilde{T}_{i,t}$ times before a decision to keep or reject a single arm is made. An apples-to-apples example comparing the two algorithms is a bit tricky due to taking different inputs (per-round budget vs. number of arms); we will add two "similar" illustrative examples in the 3-arms/2-stage case to help compare and contrast the two algorithms, though.

**Greedy subset selection (R4):** We will run against the greedy algorithm, as suggested. We find the most relevant comparison to be the previous work ("SWAP") from Schumann et. al. 2019. Our method has theoretical guarantees, as does theirs. (Note that we also compare to the actual admissions decision process, which is fairly complex.)

**Anonymized data release (R4):** Unfortunately, we are prohibited via FERPA regulations from releasing the real dataset. That said, we are working (i) with our IRB for an anonymized release option as suggested (e.g., by fitting a blanket parameterized model) and (ii) to release through our multi-university research consortium a general set of scripts that mimic our application scraping (e.g., running OCR on SoPs and rec. letters, parsing structured data such as GRE/GPA) and feed that generic data into our algorithms run on their servers. This is a longer process that will require rounds of validation, but our long-term goal is to collect aggregate statistics and push practical impact in this way.

[Meta-Review · NeurIPS 2019]

This paper proposes a novel idea of multi-armed bandit formulation to tiered interviews. While selecting top K arms is not a new problem, application to interviewing and selecting set of candidates seems novel. The technical contributions and theory seem non-trivial, sound, and feasible. There are few aspects that need to be addressed as per reviewer comments: 1. Comparing the results of the proposed algorithm to a greedy baseline 2. Improving the clarity on making a connection between tiered interviewing and the proposed bandit formulation. 3. As suggested by reviewer 1, making the experimental setup more realistic and clarifying how to interpret the comparison of the proposed approach with the baselines (which are designed with different objectives) is critical. All in all, I think this is a borderline paper with some interesting ideas.